# Differential Genetic and Epigenetic Effects of the *KLF14* Gene on Body Shape Indices and Metabolic Traits

**DOI:** 10.3390/ijms23084165

**Published:** 2022-04-09

**Authors:** Semon Wu, Lung-An Hsu, Ming-Sheng Teng, Hsin-Hua Chou, Yu-Lin Ko

**Affiliations:** 1Department of Life Science, Chinese Culture University, Taipei 11114, Taiwan; semonwu@yahoo.com.tw; 2The First Cardiovascular Division, Department of Internal Medicine, Chang Gung Memorial Hospital and Chang Gung University College of Medicine, Taoyuan 33305, Taiwan; hsula@adm.cgmh.org.tw; 3Department of Research, Taipei Tzu Chi Hospital, Buddhist Tzu Chi Medical Foundation, New Taipei City 23142, Taiwan; vincent@tzuchi.com.tw; 4The Division of Cardiology, Department of Internal Medicine and Cardiovascular Center, Taipei Tzu Chi Hospital, Buddhist Tzu Chi Medical Foundation, New Taipei City 23142, Taiwan; chouhhtw@gmail.com; 5School of Medicine, Tzu Chi University, Hualien 97004, Taiwan

**Keywords:** *KLF14*, body shape indices, metabolic traits, differential effect, genetic variants, DNA methylation

## Abstract

The *KLF14* gene is a key metabolic transcriptional transregulator with monoallelic maternal expression. *KLF14* variants are only associated with adipose tissue gene expression, and *KLF14* promoter methylation is strongly associated with age. This study investigated whether age, sex, and obesity mediate the effects of *KLF14* variants and DNA methylation status on body shape indices and metabolic traits. In total, the data of 78,742 and 1636 participants from the Taiwan Biobank were included in the regional plot association analysis for *KLF14* variants and *KLF14* methylation, respectively. Regional plot association studies revealed that the *KLF14* rs4731702 variant and the nearby strong linkage disequilibrium polymorphisms were the lead variants for lipid profiles, blood pressure status, insulin resistance surrogate markers, and metabolic syndrome mainly in female participants and for body shape indices mainly in obese women. Significant age-dependent associations between *KLF14* promoter methylation levels and body shape indices, and metabolic traits were also noted predominantly in female participants. *KLF14* variants and *KLF14* hypermethylation status were associated with metabolically healthy and unhealthy phenotypes, respectively, in obese individuals, and only the *KLF14* variants demonstrated a significant association with both higher adiposity and lower cardiometabolic risk in the same allele, revealing uncoupled excessive adiposity from its cardiometabolic comorbidities, especially in obese women. Variations of *KLF14* are associated with body shape indices, metabolic traits, insulin resistance, and metabolically healthy status. Differential genetic and epigenetic effects of *KLF14* are age-, sex- and obesity-dependent. These results provided a personalized reference for the management of cardiometabolic diseases in precision medicine.

## 1. Introduction

Krüpple-like factors (KLFs), belonging to the SP/KLF family, are a group of evolutionarily conserved transcription factors with C-terminal zinc finger domains that are crucial for the recognition of and binding to the GT/GC-rich or CACCC box cis-regulatory sites in gene promoters and enhancers [1,2]. KLFs play a critical role in homeostasis maintenance by regulating the gene expression in many organ systems, such as cardiovascular, renal, digestive, respiratory, immunological, and hematopoietic systems [3,4,5,6]. KLFs also regulate cell signaling pathways controlling cell proliferation, apoptosis, migration, differentiation, and other physiological activities [4]. As a member of the KLF family, KLF14 is induced by TGF-β in intraembryonic and ectodermal tissues [4,6] and forms a corepressor transcription complex with Sin3A and HDAC2 to inhibit the expression of TGF-II receptors. KLF14 is only expressed in mammalian tissues, including the muscle, brain, fat, and liver, and it has monoallelic maternal expression in all tissues of both humans and mice [7]. In the livers of high-fat-diet–fed mice and db/db obese mice, KLF14 activates the PI3K/Akt signaling pathway to increase insulin sensitivity [8]. It also mediates lipid signaling, and KLF14-mediated metabolic phenotypes are attributed to the regulation of lipid signaling through the dysregulation of sphingokinase 1, a critical lipid signaling molecule [5,9,10]. With the lack of *Klf14*-mediated apoA-I promoter activation, hepatic-specific *Klf14* deletions in *Apoe^−/−^* mice cause marked acceleration of atherosclerotic lesion development in combination with decreased high-density lipoprotein (HDL) cholesterol levels, and cholesterol efflux, whereas Klf14 activation reduces atherosclerosis [11]. Together, these findings indicate that epigenetic regulation–induced *KLF14* loss-of-function may trigger the onset of metabolic diseases. Furthermore, KLF14 is a master metabolic transcriptional regulator that mediates adipogenesis, insulin signaling, lipid metabolism, inflammatory and immune responses, and cell proliferation and differentiation, thus acting as a potential novel therapeutic target to reduce the risk of atherosclerotic cardiovascular disease [12].

The *KLF14* gene is a single exon imprinted gene localized in the human chromosome 7q32.2, with a total length of 1059 bp; it encodes 323 amino acids, and only the allele inherited from the mother is expressed [13]. Using chromatin-state maps, a 1.6-kb enhancer region approximately 5 kb upstream of the *KLF14* transcription start site has been identified in adipose tissue [14]. Genome-wide association studies (GWASs) have identified *KLF14* variants to be associated with a multitude of metabolic pathologies, such as insulin resistance (IR), diabetes mellitus, coronary artery disease, ischemic stroke, and myocardial infarction [14,15,16,17]. Lead single-nucleotide polymorphisms (SNPs) in GWASs map to a 3–48-kb region upstream of *KLF14*, and these mapped genetic variants are also associated with the abundance of the *KLF14* transcript only in adipose tissue [14,18]. The associations are usually stronger in women than in men. Thus, the potential interactive effect of sex and obesity on the association between *KLF14* variants and cardiometabolic phenotypes required further elucidation.

Aging is an important risk factor for chronic metabolic and inflammatory disorders such as atherosclerosis, cancer, and type 2 diabetes mellitus [19,20,21]. CpG sites located in *KLF14* are associated with aging-related alteration of hypermethylation in whole blood samples, thus providing an accurate estimate of chronological age and serving as age-related epigenetic biomarkers [22,23,24]. In addition, the age-related DNA methylation level of *KLF14* may be a risk marker for diabetes mellitus [25]. Recently, extensive studies have also focused on the genetic loci of metabolically healthy phenotype in obese individuals and of genetic loci that uncouple excessive adiposity from its cardiometabolic comorbidities [26,27]. The Taiwan Biobank (TWB) population-based cohort study enrolled >100,000 volunteers aged 30–70 years with no history of cancer [28,29]. In the current study, we included participants from the TWB cohort study to elucidate the effects of age, sex, and obesity on the association of *KLF14* variants and methylation status with conventional and allometric body shape indices, IR surrogate marker levels, various metabolic traits, and metabolic syndrome. We also assessed whether *KLF14* variants and methylation levels are markers of metabolically healthy obese phenotype and whether these variations uncouple excessive adiposity from the adverse metabolic traits in Taiwan.

## 2. Results

### 2.1. Regional Plot Association Studies for Genetic Variants at Positions between 130.3 and 130.5 Mb on Chromosome 7q32.2 and Study Phenotypes

The flowchart of participant enrollment is presented in Figure 1. We performed regional plot association studies in 78,742 TWB participants to examine the association between genetic variants at positions between 130.3 and 130.5 Mb on chromosome 7q32.2 and the studied phenotypes. Lead SNPs with genome-wide significance were noted for the HDL cholesterol level (rs4731702, *p* = 6.69 × 10^−14^), triglyceride level (rs13240528, *p* = 1.46 × 10^−16^), metabolic syndrome (rs3996352, *p* = 1.30 × 10^−13^), mean blood pressure (BP) (rs1364422, *p* = 3.59 × 10^−8^), and hip index (HI) (rs35057928, *p* = 3.48 × 10^−14^), whereas associations with *p* < 1.41 × 10^−4^ were noted for body mass index (BMI) (rs3996352, *p* = 1.58 × 10^−6^), a body shape index (ABSI) (rs34072724, *p* = 2.14 × 10^−6^), hip circumference (rs35057928, *p* = 3.54 × 10^−7^), and waist circumference (rs34072724, *p* = 1.41 × 10^−5^) (Table 1 and Appendix A). Except for rs1364422, which is the lead SNP for mean BP, all other lead SNPs were in nearly complete LD with rs4731702 (r^2^ > 0.95) (Appendix A). Our data revealed that the lead SNP for each phenotype was located upstream of the *KLF14* gene region, revealing pleiotropic effects on this gene locus.

### 2.2. Association of the KLF14 rs4731702 Genotype with Parameters of Body Shape Indices, IR Surrogate Markers, and Metabolic Traits

Because of nearly complete LD between lead *KLF14* polymorphisms, we selected *rs4731702*, a polymorphism associated with adipocyte size and body composition traits on *KLF14* [14], for further genotype–phenotype analysis of the various clinical phenotypes and laboratory parameters of 78,742 participants. By using an additive model, after adjustment for age, sex, BMI, and smoking status, we observed genome-wide significant associations of *rs4731702* with HI, HDL cholesterol, and triglyceride levels and metabolic syndrome, whereas associations with *p* < 2.65 × 10^−4^ were noted for hip circumference, waist circumference, body fat percentage (BFP), BMI, ABSI, systolic BP, diastolic BP, and mean BP. The products of triglyceride (TG) and fasting plasma glucose (the TyG index), TyG and body mass index (TyG-BMI), TyG and waist circumference (TyG-WC), and hypertension (Appendix A) influence type 2 diabetes risk via a female-specific effect on adipocyte size and body composition.

### 2.3. Interactive Effects of Sex and Obesity on the Association between the KLF14 rs4731702 Genotype and Various Phenotypes

We investigated whether sex and obesity affect the association between the *rs4731702* genotype and the studied phenotypes. As presented in Table 2, for female participants, genome-wide significant associations were found between the *rs4731702* genotype and hip circumference, HI, HDL cholesterol and triglyceride levels, TyG index, TyG-BMI, and metabolic syndrome. Significant associations with *p* < 2.65 × 10^−4^ were found between the *rs4731702* genotype and waist circumference, BMI, ABSI, diastolic and mean BP, TyG-WC, and diabetes mellitus in women. However, no significant associations were found between the *rs4731702* genotype and the studied phenotypes in men. The results of the two-sample *t* tests revealed that sex affected the association between the *KLF14* rs4731702 genotype and waist circumference, hip circumference, ABSI, HI, HDL cholesterol, and triglyceride levels (Table 2). These results suggest that the association between the *rs4731702* genotype and several phenotypes is sex dependent. We further evaluated the genotype–phenotype associations of *rs4731702* in obese and nonobese participants. Differential associations according to adiposity status were noted only in body shape indices, including waist and hip circumferences and ABSI, with significant associations occurring only in obese participants but not in nonobese participants (Appendix A). We then divided the participants into four groups according to sex and obesity. No obvious associations were noted between the *rs4731702* genotype and various phenotypes in men, whether obese or nonobese (Appendix A). In women, differential associations with adiposity status were noted between the *rs4731702* genotype and body shape indices but not between the *rs4731702* genotype and metabolic traits (Table 3).

### 2.4. Association between KLF14 Promoter Methylation Status and KLF14 Variants and Age

We examined whether *KLF14* variants are associated with the DNA methylation status of the *KLF14* promoter region. We observed no significant association between *KLF14* variants and methylation levels in 32 *KLF14* methylation sites (Appendix A). By contrast, after adjustment for sex, BMI, and smoking, genome-wide significant associations were observed between age and 15 *KLF14* promoter DNA methylation sites, with hypermethylation associated with increasing age and with the minimum *p* value of 3.83 × 10^−253^ for *KLF14* cg08097417 (Figure 2). Both male and female participants exhibited a highly significant association between age and *KLF14* promoter methylation status (Appendix A).

### 2.5. Associations between KLF14 Methylation Status at the Promoter Region and Various Phenotypes Are Mediated by Chronologic Age

We evaluated the association between a *KLF14* promoter methylation site, cg08097417, which displayed the strongest association with age and various phenotypes. After adjustment for sex, BMI, and current smoking, our data revealed genome-wide significant positive associations between cg08097417 methylation and various body shape indices and metabolic traits, such as waist circumference, waist–hip ratio, ABSI, waist–hip index (WHI), systolic and mean BP, total cholesterol levels, and hypertension. Negative associations were observed in cg08097417 methylation with body height, body weight, and hip circumference, and positive associations with *p* < 2.65 × 10^−4^ between cg08097417 methylation and diastolic BP, low-density lipoprotein (LDL) cholesterol, triglyceride, and hemoglobin A1c (HbA1c) levels and diabetes mellitus, which are in the same direction with age-related adiposity and metabolic changes (Table 4). In addition, after further adjustment for age, significance levels markedly decreased, and all the associations became nonsignificant, suggesting that the associations are predominantly mediated by chronologic age (Table 4). We further extended the analysis to other *KLF14* promoter methylation sites with regional plot association studies. The results revealed similar association trends between the *KLF14* promoter methylation levels and body shape indices and metabolic traits, with subsidence of significant associations after adjustment for age (Figure 3).

### 2.6. Associations between KLF14 Methylation Status at the Promoter Region and Various Phenotypes: Subgroup Analysis with Sex and Obesity

Subgroup analysis revealed an interactive effect of sex on the associations between cg08097417 methylation levels and several metabolic traits, including systolic BP, total and LDL cholesterol and triglyceride levels, and HbA1c, with the association occurring predominantly in the female sex (Appendix A and Figure 4). By contrast, no interactive effect was noted on the associations between cg08097417 methylation levels and all study phenotypes according to adiposity status (Appendix A).

### 2.7. KLF14 rs4731702 Variant as a Marker of Metabolically Healthy Obese Phenotype with the Adiposity-Increasing Allele Associated with a Favorable Cardiometabolic Risk Profile

We further evaluated whether the *KLF14* rs4731702 genotypes and *KLF14* promoter methylation levels are involved in the uncoupling of adiposity from its adverse metabolic risk profiles using the standardized effect size analysis in the Radar plot. From the Radar plots of different subgroups of participants, the *KLF14* rs4731702 C allele was the most strongly and positively associated with HI, ABSI, hip and waist circumferences, BMI, and BFP, but no association with waist–hip ratio and WHI was found (Figure 5A–C). By contrast, the *KLF14* rs4731702 C allele was negatively associated with nearly all metabolic traits, positively associated with HDL cholesterol levels, and had no association with total and LDL cholesterol levels. These results were provided by analyses for total participants, female participants, and obese female participants (Figure 5A–C). By contrast, cg08097417 hypermethylation showed a positive association with both body shape indices and metabolic traits, except for body height, hip circumference, and HI (Figure 5D). In the analysis conducted using the criteria of Park et al. [27], our data indicated that the rs4731702 C allele has a lower risk of metabolically unhealthy status compared with metabolically healthy status in obese participants (odds ratio: 0.88, 95% confidence interval [CI]: 0.85–0.92, *p* = 1.92 × 10^−11^), whereas hypermethylation of the cg08097417 methylation site was associated with a higher risk of metabolically unhealthy status in obese participants (odds ratio 4.04, 95% CI 1.95–8.41, *p* = 1.81 × 10^−4^) (Appendix A).

## 3. Discussion

This study investigated the differential genetic and epigenetic effects of *KLF14* on the association between body shape indices and metabolic traits. Several novel results were identified. First, we observed an independent association of genetic and epigenetic effects of *KLF14* on both multiple body shape indices and metabolic traits. Second, the association of *KLF14* variants with body shape indices and metabolic traits occurred predominantly in the female sex, and the association of *KLF14* variants with body shape indices occurred predominantly in obese participants. Thus, obese women were the only subgroup associated with both body shape indices and metabolic traits. Third, *KLF14* methylation is a marker of age in forensic medicine for both sexes. In addition to diabetes mellitus, *KLF14* promoter methylation status showed a significant age-dependent association with various body shape indices and metabolic traits, and the associations with lipid profiles, HbA1c, and metabolic syndrome occurred predominantly in female participants. Finally, differential genetic and epigenetic effects were noted in cross-phenotype associations using the adiposity–cardiometabolic trait pairs. *KLF14* rs4731702 variant, but not methylation status, served as a marker involved in the uncoupling of adiposity from its adverse metabolic risk profiles predominantly in subgroups of obese female participants. To the best of our knowledge, this is the first report investigating the mediated and interactive effects of age, sex, and obesity on the association of genetic and epigenetic *KLF14* variations with body shape indices and metabolic traits. These results support the critical role of *KLF14* as an age-, sex-, and obesity-specific key transcriptional regulator affecting a large transregulatory network of metabolic traits and adiposity status, and the results also provided evidence for the differential genetic and epigenetic effects of *KLF14* on the risk of cardiometabolic disorders.

### 3.1. Association between KLF14 Variants and Metabolic Traits: The Role of Sex and Obesity

By including the *KLF14* rs4731702 genotypes in the analysis, Small et al. [14] demonstrated colocalization of the GWAS signal for metabolic disorders and the expression quantitative trait locus (eQTL) for nearly 400 genes in trans to the *KLF14* locus. The results revealed *KLF14* to be one of the largest trans-eQTL hotspots in the human genome [30]. Despite the nearly ubiquitous expression of KLF14 in many tissues [31], *KLF14* variants regulate gene expression only in adipose tissue [14]. *KLF14* is also an imprinted gene with monoallelic maternal expression [14]. Our data indicated a significant association of *KLF14* variants with various metabolic traits, such as lipid profiles, BP status, IR surrogate marker levels, and metabolic syndrome, but only in female participants. By contrast, no interactive effect for such associations according to adiposity status was noted in our study population.

### 3.2. Association of KLF14 Variants with Adiposity Indices, including Body Shape and Body Fat Distribution

Body shape plays a critical role in influencing cardiometabolic complications of obesity, with the abdominal size showing positive associations and gluteofemoral size showing inverse associations with complications [32]. Conventional body shape indices, such as WHR and waist and hip circumferences, are strongly associated with BMI, and even after adjustment for BMI, waist and hip circumferences are strongly associated with height [14]. Allometric body shape indices, such as ABSI and HI, are alternatives to body shape indices and are independent of body size and general obesity [33,34]. In analogy to ABSI as an allometric index of waist circumference, WHI was created as an allometric index of WHR [35]. ABSI has been shown to be significantly associated with cardiometabolic risk factors and mortality [33,36]. ABSI also achieves better mortality risk stratification than other body shape indices, which are correlated with BMI [37]. Similar to our results with sexual dimorphism, GWAS of allometric body shape indices in the UK Biobank Resources of 406,697 participants with British ancestry indicated a genome-wide significant association of *KLF14* variants with HI in women but not in men [35]. We further revealed the interactive effect of obesity on the association between *KLF14* variants and body shape indices. In subgroup analysis, the association between *KLF14* variants and body shape indices occurred only in obese women, showing the critical role of both sex and obesity in associations between *KLF14* variants and body shape indices.

### 3.3. Epigenetic Effects of KLF14 on Adiposity Indices and Metabolic Traits Occur Only in the Female Sex and Are Dependent on Age

Aging generally induces global hypomethylation with a genome-wide decrease in the average DNA methylation level. By contrast, aging-related hypermethylation events occurred in 13% of the CpG sites among the millions of sites in the genome [38]. *KLF14* has a large CpG island that spans almost its entire open reading frame. By analyzing KLF14 in offspring from a DNA methyltransferase 3a conditional knockout mouse, *KLF14* expression was detected within the hypermethylated region inherited from the mother, which is a typical feature of the KLF family [7]. Using pathway analyses of adipose tissue from 190 individuals, altered DNA methylation in 1050 genes, including *KLF14*, has been reported as epigenetic markers of chronological age in blood [39]. DNA methylation in *KLF14* is linearly correlated with human age with an increase in the methylation level of the *KLF14* promoter region in multiple tissues in mice, such as the whole blood, adipose tissue, kidney, spleen, lung, and colon, which is significantly associated with inflammatory marker levels and with decreased *KLF14* downstream gene methylation in the whole blood, the adipose tissue, and the spleen [40]. More importantly, DNA methylation loci in *KLF14* is an age-related epigenetic biomarker; it can be used as an age prediction model to predict chronological age and may be considered an important factor in the development of aging-associated diseases, such as cancer, diabetes mellitus, and cardiovascular diseases [38,41].

Our data revealed the hypermethylation of the *KLF14* promoter region with age in both sexes. We further demonstrated significant associations of *KLF14* promoter methylation with body shape indices and metabolic traits, and the association occurred predominantly in the female sex, which was similar to the results of *KLF14* variants. However, no significant association between *KLF14* variants and *KLF14* promoter methylation was noted. These results are consistent with those of a previous study reporting a significant association between the *KLF14* rs4731702 genotype and *KLF14* gene expression and DNA methylation in adipose tissue, but not in whole blood. The finding suggested that as a blood biomarker, *KLF14* exerts both genetic and epigenetic effects on body shape indices and metabolic traits independently [14]. The association of *KLF14* methylation with various phenotypes was abolished after adjustment for age, revealing an age-dependent effect. Thus, *KLF14* methylation can serve not only as an epigenetic biomarker of age but also as an epigenetic biomarker of age-related changes in adiposity status, body shape indices, and metabolic phenotypes.

### 3.4. KLF14 rs4731702 Variant as a Marker of Metabolically Healthy Obese Phenotype That Uncouples Excessive Adiposity from Its Comorbidities

Obesity is typically associated with chronic, low-grade inflammation and a constellation of metabolic abnormalities, which are important risk factors for type 2 diabetes and cardiovascular diseases [42,43]. For a given fat mass, individuals who are obese may not develop metabolic comorbidities, which was termed metabolically healthy obesity (MHO). The mechanism of the discrepancy has been suggested to be related to disproportionate adipose tissue distribution in metabolically unhealthy obesity (MUO) with visceral adiposity [44,45] or an impaired ability to expand subcutaneous fat in the lower part of the body. Pathway analyses have suggested that pathophysiological mechanisms for the differential status of metabolically healthy or unhealthy individuals may involve adipogenesis, angiogenesis, transcriptional regulation, and IR [46]. Cumulative evidence from several large-scale prospective studies suggests that MHO individuals have higher risks than healthy normal-weight individuals of many cardiometabolic outcomes and cardiovascular diseases, such as hypertension, IR, diabetes mellitus, and cerebrovascular disease [47,48,49,50,51,52,53,54,55,56]; yet, their risk is lower than that of MUO individuals [46].

Recently, GWASs have identified genetic variants that are associated with increased adiposity in conjunction with a healthy metabolic profile [46]. GWAS investigating genetic determinants of insulin concentrations in individuals without diabetes highlighted that many identified loci showed associations with markers suggestive of clinical IR, such as high triglyceride and low HDL concentrations [57]. Favorable adiposity alleles were also associated with a higher subcutaneous fat mass and lower liver fat content, supporting the hypothesis that storing excess triglycerides in metabolically low-risk depots has beneficial effects on metabolism [57]. Waist-specific and hip-specific polygenic risk scores, which are related to visceral and subcutaneous abdominal fat and gluteofemoral and leg fat, respectively, increased the risks of an adverse metabolic profile, type 2 diabetes, and coronary artery disease [58]. In a genome-wide discovery of genetic loci that uncouple excess adiposity from its comorbidities, Huang et al. [26] identified 62 loci with the same allele significantly associated with both higher adiposity and lower cardiometabolic risk. These loci are enriched for genes expressed in adipose tissue and for regulatory variants influencing nearby genes affecting adipocyte differentiation. In a Korean study examining the genetic determinants of MHO and metabolically unhealthy normal weight phenotypes, the genes common to both models are related to lipid metabolism, and those associated with MHO are related to insulin or glucose metabolism [27]. Our data indicated an inverse association between the rs4731702 genotypes and multiple body shape indices when compared with the association with metabolic traits and IR surrogate markers. These results supported *KLF14* variants as a marker for uncoupling excessive adiposity from its cardiometabolic risk profiles, which occurred predominantly in obese women. By contrast, *KLF14* hypermethylation is associated with a metabolically unhealthy obese phenotype coupling of adiposity with its comorbidities. Interestingly, Cannataro et al. [59] found that microRNAs may modify promoter by methylation on targeted genes and regulate their expression, which affect body shape under Ketogenic Diet. Therefore, we definitely consider investigating the microRNA candidates that regulate KLF14 associated body shape in the near future.

### 3.5. Limitation

This study has limitations. First, ethnic genetic heterogeneity has been noted in the genetic association studies, and our data may not be applied to other ethnic populations. Second, several regional plot association studies did not reach genome-wide significance for the associations. Further studies with larger sample size and using different ethnic populations may help to strengthen the validity of our analysis.

## 4. Participants and Methods

### 4.1. TWB Participants

We included 107,494 TWB participants with no history of cancer who had undergone genotyping from Axiom Genome-Wide CHB 1 or 2 Array (Affymetrix, Santa Clara, CA, USA). Of them, 28,752 were excluded because of the absence of imputation data (12,289 participants) and after quality control (QC) for the GWAS with identity by descent (IBD) > 0.187 revealing relative pairs of second-degree relatives or closer (10,956 participants). Participants with fasting <6 h (2862 participants) and the absence of any of the study phenotypes (2645 participants) were also excluded. When examining the lipid profile, blood pressure status, and glucose metabolism parameters, the participants with a history of hyperlipidemia, hypertension, and diabetes mellitus were also excluded. The flowchart of participant enrollment is presented in Figure 1. The definitions of hypertension, diabetes mellitus, obesity, current smoking, and metabolic syndrome are provided in Appendix A. Ethical approval was received from the Research Ethics Committee of Taipei Tzu Chi Hospital, Buddhist Tzu Chi Medical Foundation (approval number: 05-X04-007), and Ethics and Governance Council of the Taiwan Biobank (approval number: TWBR10507-02 and TWBR10611-03). All participants signed an informed consent form before participating in the study.

### 4.2. Clinical Phenotypes and Laboratory Examinations

We measured the following clinical parameters: body height, body weight, and conventional body shape indices such as waist circumference, hip circumference, waist–hip ratio, and BMI, mean heart rate, and systolic, mean, and diastolic BP. We also collected the following biochemistry data: lipid profiles and glucose metabolism parameters such as total HDL and LDL cholesterol and triglyceride levels, fasting plasma glucose levels, and HbA1c. TyG index, TyG-BMI, and TyG-WC were used as IR surrogate markers [60]. BFP was measured using the Body Composition Analyzer BC-420MA (TANITA, TANITA Corporation, Sportlife Tokyo, Japan), a device that sends a weak electric current through the body to measure the impedance (electrical resistance) of the body. ABSI, WHI, and HI were calculated as parameters of allometric body shape indices [35]. The measurement details of BFP and the calculation of conventional and allometric body shape indices and TyG-related parameters are provided in Appendix A. We also performed cross-phenotype analysis for *KLF14* variants and *KLF14* methylation levels and subsequently calculated the standard deviation effect sizes *(β)* of their effects on the body shape indices and metabolic traits to elucidate whether *KLF14* variations may uncouple excessive adiposity from its comorbidities [26]. The resulting effect sizes are on the same scale, making them comparable across all variants, methylation levels, and traits. By using the criteria of Park et al. [27], we further examined the association of *KLF14* variants and *KLF14* methylation status with metabolically healthy and unhealthy obese phenotypes in TWB participants.

### 4.3. DNA Isolation and Genotyping of KLF14 Variants

DNA was isolated from blood samples using a PerkinElmer Chemagic 360 instrument (PerkinElmer, Waltham, MA, USA). SNP genotyping was conducted using custom TWB chips and was performed on the Axiom genome-wide array plate system (Affymetrix, Santa Clara, CA, USA).

### 4.4. Regional Plot Association Studies

To determine the lead SNPs around the *KLF14* region for various studied phenotypes, we performed a QC for regional plot association analysis by including TWB participants enrolled after applying the exclusion criteria (Figure 1). The Axiom genome-wide CHB 1 and 2 array plates, each with 27,720 and 79,774 participants and comprising 611,656 and 640,160 SNPs, respectively, were used for imputation analysis. Genome-wide genotype imputation was performed using SHAPEIT and IMPUTE2, with the data of the East Asian population from the 1000 Genome Project Phase study used as the reference panel. After imputation, indels were removed using VCFtools, and the QC was performed by filtering SNPs with an IMPUTE2 imputation quality score of >0.3. All samples enrolled for the analysis had the identity by descent PI_HAT > 0.1875. For SNP QC, three criteria were used for exclusion from subsequent analyses: (1) an SNP call rate of <3%, (2) a minor allele frequency of <0.01, and (3) the violation of the Hardy–Weinberg equilibrium with *p* < 10^−6^. After QC, the data of 85,508 participants and 131 SNPs at positions between 130.3 and 130.5 Mb on chromosome 7q32.2 were included in the regional plot association analysis.

### 4.5. DNA Methylation Analysis

DNA methylation was assessed using sodium-bisulfite-treated DNA from whole blood using the Infinium MethylationEPIC BeadChipEPIC array (Illumina, San Diego, CA, USA). Four criteria were used to perform the QC for DNA methylation analysis: (1) normalization across batches for correction of dye bias, (2) background signal removal, (3) using the median absolute deviation method to eliminate outliers, and (4) elimination of probes with poor detection (*p* > 0.05) and those whose bead counts were <3. Finally, the data of 1702 participants were included in the *KLF14* methylation analysis. Sixty-six participants were further excluded due to fasting for <6 h (*n* = 16) or the absence of any study phenotype (*n* = 50).

### 4.6. Statistical Analysis

Continuous variables were expressed as median and interquartile ranges when the distribution was strongly skewed. Categorical data distributions were expressed as a percentage. Before analysis, all study parameters with a skew distribution were logarithmically transformed to adhere to the normality assumption. To test for significant differences in the association between different subgroups of sex and obesity, we used a two-sample *t* test [32]. We assumed the genetic effect to be additive, and a general linear model was used to analyze the studied phenotypes in relation to the predictors of investigated genotypes and confounders. Multivariable logistic regression analysis was used to evaluate the independent effect of genotypes on the risk of hypertension, diabetes mellitus, and metabolic syndrome. The Radar plots were presented with the standardized coefficients (beta) in the linear regression. Genome-wide scans were performed, and the beta coefficient, standard error, and *p* values by general linear model or logistic regression for genome-wide data were analyzed using the analysis software package PLINK. Genome-wide significance was defined as a *p* value of <5 × 10^−8^. For Bonferroni correction of regional plot association studies, the significant value was defined as *p* < 1.41 × 10^−5^ calculated as 0.05/(131 × 27) according to a total of 131 variants and 27 traits analyzed. For Bonferroni correction of each genotype–phenotype and methylation–phenotype analysis, the significant value was defined as *p* < 2.65 × 10^−4^, calculated as 0.05/(27 × 7) according to 27 traits and 7 subgroups analyzed. Linkage disequilibrium (LD) between each SNP was analyzed using LDmatrix software (https://analysistools.nci.nih.gov/LDlink/?tab=ldmatrix, accessed on 19 July 2021). All statistical analyses were performed using SPSS (version 22; SPSS, IBM Corp., Armonk, NY, USA) or PLINK. Missing data were approached with listwise deletion.

## 5. Conclusions

Taken together, our data revealed that variations of the *KLF14* gene, including gene variants and methylation levels for body shape indices, IR status, and metabolic traits, are highly dependent on age, sex, or obesity. These results support the critical role of *KLF14* as a key age-, sex-, and obesity-specific transcriptional regulator affecting a large adipose-specific transregulatory network of metabolic traits and adiposity status, which provide further evidence for the differential genetic and epigenetic effects of *KLF14* on the risk of cardiometabolic disorders.

## Figures and Tables

**Figure 1 ijms-23-04165-f001:**
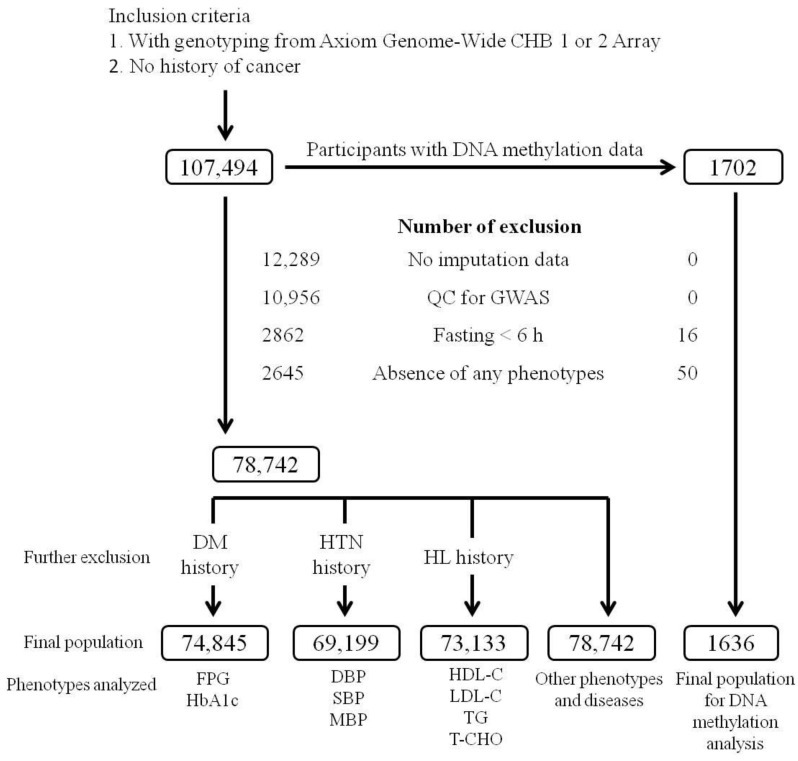
Study inclusion and exclusion criteria flowchart. This flowchart presents the inclusion and exclusion criteria used to screen for Taiwan Biobank (TWB) participants. GWAS—genome-wide association study; QC—quality control; HL—hyperlipidemia, HTN—hypertension; DM—diabetes mellitus; FPG—fasting plasma glucose; HbA1c—glycated hemoglobin; SBP—systolic blood pressure; DBP—diastolic blood pressure; MBP—mean blood pressure; LDL-C:—low-density lipoprotein cholesterol; HDL-C—high-density lipoprotein cholesterol; TG—triglyceride; T-CHO—total cholesterol. Other phenotypes include age, body mass index (BMI), waist circumference, hip circumference, waist–hip ratio, a body shape index (ABSI), waist–hip index (WHI), hip index (HI), the product of triglyceride and fasting plasma glucose (the TyG index), TyG with adiposity status (TyG-BMI), and TyG with waist circumference (TyG-WC), current smoking status, and metabolic syndrome.

**Figure 2 ijms-23-04165-f002:**
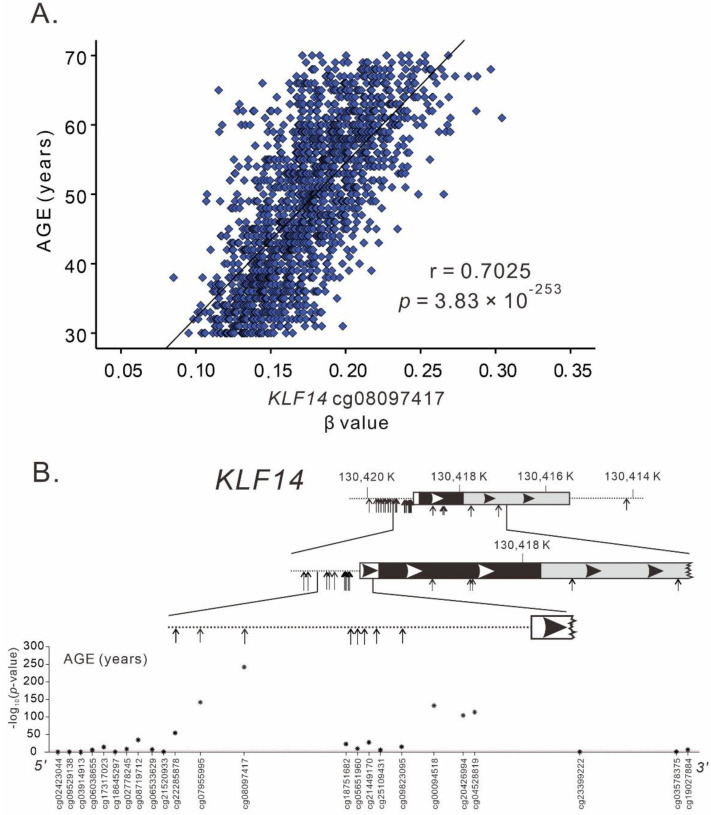
*KLF14* methylation and age: (**A**) Association between *KLF14* promoter methylation cg08097417 level and chronologic age and (**B**) Genomic structure of *KLF14* and the association between *KLF14* methylation status and age.

**Figure 3 ijms-23-04165-f003:**
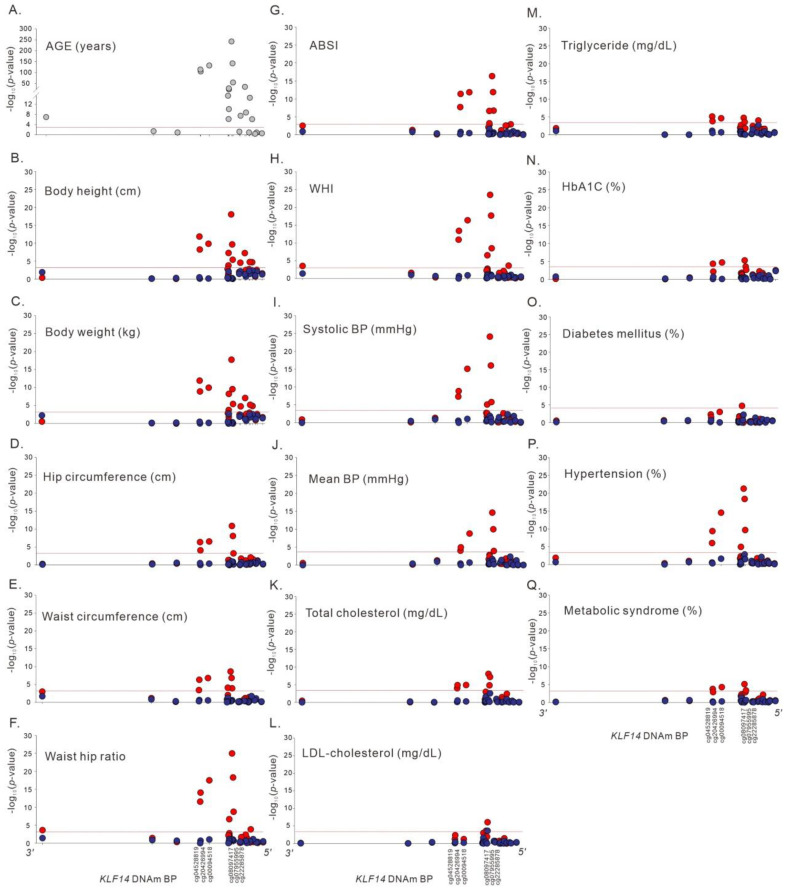
Regional plot association analysis between*KLF14* methylation status and the studied phenotypes (**A**) Age, (**B**) Body height, (**C**) Body weight, (**D**) Hip circumference, (**E**) Waist circumference, (**F**) Waist hip ratio, (**G**) ABSI, (**H**) WHI, (**I**) Systolic BP, (**J**) Mean BP, (**K**) Total cholesterol, (**L**) LDL-cholesterol, (**M**) Triglyceride, (**N**) HbA1c, (**O**) Diabetes mellitus, (**P**) Hypertension, (**Q**) Metabolic syndrome, with (blue circle) or without (red circle) adjustment for age. The results showed subsidence of significant associations after adjustment for age.

**Figure 4 ijms-23-04165-f004:**
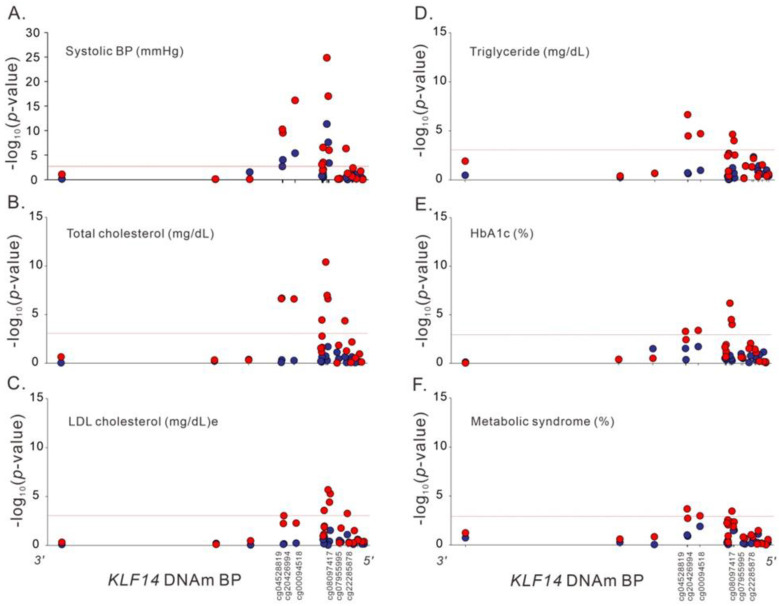
Regional plot association analysis between the studied phenotypes (**A**) Systolic BP, (**B**) Total cholesterol, (**C**) LDL cholesterol, (**D**) Triglyceride, (**E**) HbA1c, (**F**) Metabolic syndrome, and DNA methylation status according to sex, with red circles representing female participants and blue circles representing male participants.

**Figure 5 ijms-23-04165-f005:**
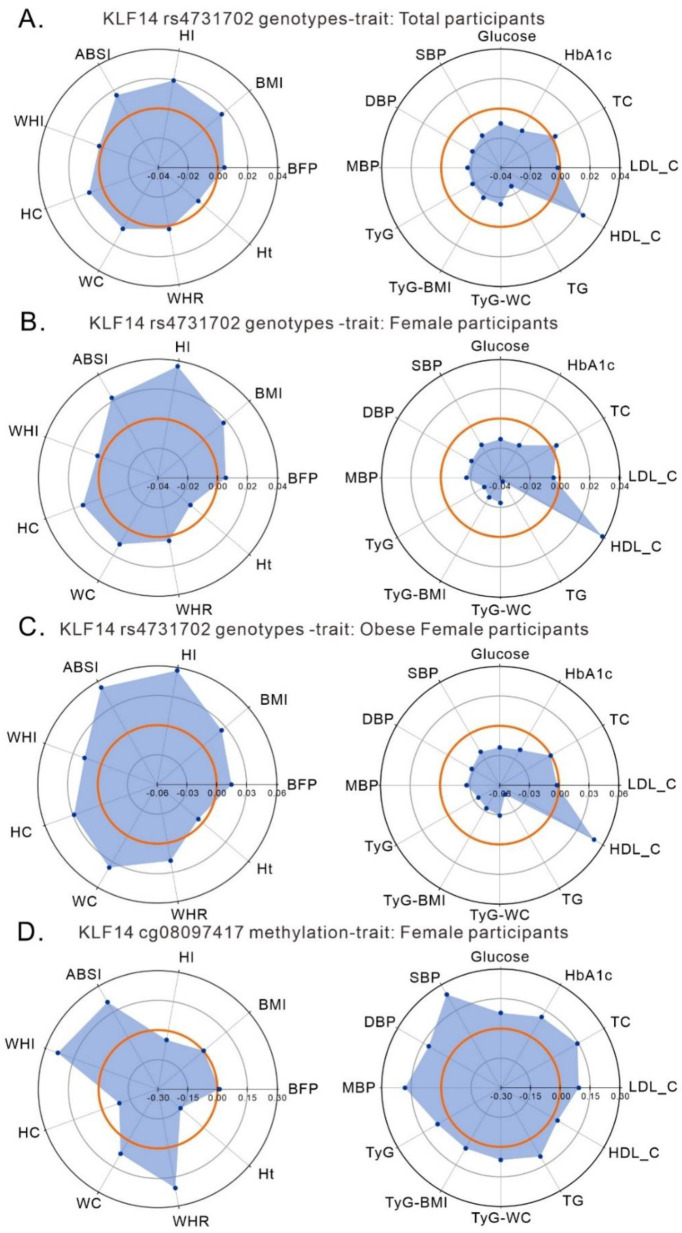
*KLF14* rs4731702 genotype (**A**–**C**) and *KLF14* promoter methylation site cg08097417 (**D**) and their associations with body shape indices (left panel) and metabolic traits (right panel) in the Taiwan Biobank population. Standardized coefficients (beta) are shown in Radar plots for total participants (**A**), female participants (**B**,**D**), and obese female participants (**C**). The middle of the three concentric circles is labeled “0,” representing no association between the *KLF14* variant and methylation status and trait. Points falling outside the middle octagon or dodecagon represent positive variant–trait or methylation–trait associations, whereas those inside it represent negative variant–trait or methylation–trait associations.

**Table 1 ijms-23-04165-t001:** Lead single nucleotide polymorphisms (SNPs) for various phenotypes at the *KLF14* gene region.

Phenotypes	Lead SNPs	Position	Ref/Alt	MAF	LD *	*p* Value
HDL cholesterol (mmol/L)	rs4731702	130748625	T/C	0.3117	1.0	6.69 × 10^−14^
Triglyceride (mmol/L)	rs13240528	130760369	A/C	0.3097	0.955	1.46 × 10^−16^
Mean blood pressure (mmHg)	rs1364422	130761222	T/C	0.4218	0.283	3.59 × 10^−8^
Metabolic syndrome (%)	rs3996352	130760175	G/A	0.3098	0.955	1.3 × 10^−13^
Body mass index (kg/m^2^)	rs3996352	130760175	G/A	0.3098	0.955	1.58 × 10^−6^
Waist circumference (cm)	rs34072724	130747710	A/G	0.3118	1.0	1.41 × 10^−5^
A body shape index	rs34072724	130747710	A/G	0.3118	1.0	2.14 × 10^−6^
Hip circumference (cm)	rs35057928	130749767	C/T	0.3104	0.978	3.54 × 10^−7^
Hip index	rs35057928	130749767	C/T	0.3104	0.978	3.48 × 10^−14^
Body fat percentage (%)	rs34084575	130751643	A/-	0.3123	1.0	0.0002

HDL—high-density lipoprotein; Ref—Reference allele; Alt—Alternate allele; MAF—Minor allele frequency; * LD—Linkage disequilibrium between rs4731702 and lead SNPs.

**Table 2 ijms-23-04165-t002:** Association between *KLF14* rs4731702 genotype and body shape indices and metabolic traits according to sex.

Clinical and Laboratory Parameters	Male (*n* = 28,483)				Female (*n* = 50,259)			
	Median (IQR)	Beta	SE	*p* Value *	Median (IQR)	Beta	SE	*p* Value *
Age (years)	51.0 (40.0–59.0)	−0.0034	0.1001	0.9733	51.0 (41.0–58.0)	0.0311	0.0705	0.6596
Body shape indices								
Body height (cm)	169.5 (165.5–173.5)	0.0041	0.0524	0.9377	157.5 (153.5–161.0)	−0.0984	0.0360	0.0063
Body weight (kg)	71.7 (65.1–79.2)	0.0108	0.0457	0.8131	56.9 (51.7–63.4)	−0.0719	0.0272	0.0081
Hip circumference (cm)	97.0 (93.0–101.3)	0.0175	0.0328	0.5927	94.0 (90.0–99.0)	0.1483	0.0261	1.40 × 10^−8 ¶^
Waist circumference (cm)	87.0 (82.0–93.0)	0.0295	0.0395	0.4549	79.6 (74.0–86.0)	0.1718	0.0368	2.98 × 10^−6 ¶^
Waist–hip ratio	0.90 (0.86–0.94)	0.0001	0.0004	0.8457	0.84 (0.80–0.89)	0.0003	0.0004	0.4090
Body fat percentage (%)	22.9 (19.6–26.1)	0.0424	0.0253	0.0939	31.6 (27.6–35.8)	0.0540	0.0153	4.17 × 10^−4^
Body mass index (kg/m^2^)	25.0 (23.0–27.3)	0.0701	0.0315	0.0262	23.0 (21.0–25.5)	0.0995	0.0251	7.19 × 10^−5^
ABSI	7.8 (7.6–8.1)	0.0015	0.0032	0.6524	7.8 (7.5–8.2)	0.0177	0.0034	2.34 × 10^−7 ¶¶^
WHI	4.0 (3.9–4.1)	0.0002	0.0017	0.9198	3.8 (3.7–4.0)	0.0014	0.0018	0.4300
HI	14.6 (14.3–14.9)	0.0022	0.0041	0.5900	15.5 (15.1–15.8)	0.0303	0.0036	2.16 × 10^−17 ¶¶^
Blood pressure and heart rate								
Mean heart rate ^†^ (/min)	34.5 (32.0–38.0)	−0.0969	0.0952	0.3088	35.0 (32.5–38.0)	0.0109	0.0620	0.8599
Systolic BP ^†^ (mmHg)	121.0 (111.5–131.3)	−0.4146	0.1412	0.0033	111.0 (102.0–123.0)	−0.3622	0.1026	4.15 × 10^−4^
Diastolic BP ^†^ (mmHg)	76.5 (70.0–83.0)	−0.3409	0.0952	3.45 × 10^−4^	69.0 (62.7–76.0)	−0.2618	0.0659	7.10 × 10^−5^
Mean BP ^†^ (mmHg)	91.2 (84.3–98.7)	−0.3654	0.1028	3.78 × 10^−4^	83.1 (76.3–91.3)	−0.2953	0.0731	5.40 × 10^−5^
Lipid profiles								
Total cholesterol ^§^ (mmol/L)	190.0 (169.0–213.0)	0.0000	0.0007	0.9856	194.0 (172.0–218.0)	0.0004	0.0005	0.4241
HDL cholesterol ^§^ (mmol/L)	47.0 (40.0–54.0)	0.0002	0.0009	0.7872	57.0 (49.0–66.0)	0.0058	0.0006	9.26 × 10^−20 ¶¶^
LDL cholesterol ^§^ (mmol/L)	121.0 (101.0–142.0)	0.0004	0.0011	0.6975	118.0 (98.0–140.0)	−0.0008	0.0008	0.3344
Triglyceride ^§^ (mmol/L)	106.0 (74.0–155.0)	−0.0030	0.0021	0.1659	82.0 (59.0–118.0)	−0.0127	0.0014	4.47 × 10^−19 ¶¶^
Glucose metabolism								
Fasting plasma glucose ^‡^ (mmol/L)	94.0 (89.0–99.0)	−0.1248	0.1576	0.4284	80.0 (86.0–95.0)	−0.3007	0.0938	0.0013
HbA1c ^‡^ (%)	5.6 (5.4–5.9)	−0.0062	0.0063	0.3223	5.6 (5.4–5.8)	−0.0129	0.0037	5.53 × 10^−4^
Insulin resistance surrogate markers								
TyG index ^§,‡^	10080.0 (6887.0–14952.0)	−161.99	117.87	0.1694	7384.0 (5200.0–10878.0)	−332.47	54.04	7.71 × 10^−10^
TyG-BMI ^§,‡^ (×10^3^)	252.5 (164.3–393.0)	−4483	3138	0.1531	170.1 (113.4–266.2)	−8072	1385	5.59 × 10^−9^
TyG-WC ^§,‡^ (×10^3^)	879.6 (578.6–1353.6)	−14,439	10905	0.1855	585.6 (395.9–900.1)	−25,264	4760	1.12 × 10^−7^
Atherosclerotic risk factors								
Diabetes mellitus (%)	12.13%	−0.0362	0.0293	0.2161	7.73%	−0.0998	0.0270	2.23 × 10^−4^
Hypertension (%)	30.56%	−0.0665	0.0217	0.0021	17.22%	−0.0510	0.0198	0.0101
Metabolic syndrome (%)	26.52%	−0.0640	0.0239	0.0073	19.34%	−0.1414	0.0200	1.47 × 10^−12^

Participants were analyzed with the exclusion of those with a history of ^†^ hypertension, ^‡^ diabetes mellitus, and ^§^ hyperlipidemia. Abbreviations as in Figure 1; SE; —standard error; IQR—Inter Quartile range. Data are presented as median (interquartile range). ** p*—adjusted for age, BMI, and current smoking; Age—adjusted for BMI and current smoking; and BMI—adjusted for age and smoking. Significance was defined as a *p* value of <0.05/(131 + 27) = 3.16 × 10^−4^. *t*-test: ^¶¶^
*p* < 0.001, ^¶^
*p* < 0.01.

**Table 3 ijms-23-04165-t003:** Association between *KLF14* rs4731702 genotype and body shape indices and metabolic traits according to obesity in females.

Clinical and Laboratory Parameters	Female, Non-Obese (*n* = 35,677)				Female, Obese (*n* = 14,582)			
	Median (IQR)	Beta	SE	*p* Value *	Median (IQR)	Beta	SE	*p* Value *
Age (years)	50.0 (41.0–58.0)	0.0035	0.0834	0.9667	52.0 (43.0–59.0)	0.1340	0.1275	0.2933
Body-shape indices								
Body height (cm)	157.5 (154.0–161.5)	−0.1199	0.0427	0.0049	156.5 (153.0–160.5)	−0.0496	0.0671	0.4597
Body weight (kg)	53.9 (50.0–57.9)	−0.0817	0.0293	0.0053	67.5 (63.0–73.3)	−0.0464	0.0597	0.4371
Hip circumference (cm)	92.0 (89.0–95.0)	0.0829	0.0290	0.0043	101.0 (98.0–105.5)	0.3001	0.0551	5.23 × 10^−8^ **
Waist circumference (cm)	76.5 (72.0–81.0)	0.0506	0.0417	0.2251	89.0 (84.5–95.0)	0.4659	0.0747	4.49 × 10^−10^ **
Waist hip ratio	0.8 (0.8–0.9)	−0.0002	0.0005	0.6526	0.8 (0.8–0.9)	0.0018	0.0008	0.0198
Body fat percentage (%)	29.4 (26.2–32.2)	0.0421	0.0171	0.0138	38.4 (36.4–41.3)	0.0998	0.0281	3.83 × 10^−4^
Body mass index (kg/m^2^)	21.8 (20.3–23.2)	0.0225	0.0156	0.1486	27.2 (25.9–29.3)	0.1106	0.0364	0.0024
ABSI	7.8 (7.5–8.2)	0.0077	0.0041	0.0593	7.8 (7.5–8.1)	0.0411	0.0062	2.64 × 10^−11^ *
WHI	3.8 (3.7–4.0)	−0.0010	0.0021	0.6383	3.8 (3.7–4.0)	0.0076	0.0033	0.0209
HI	15.6 (15.2–15.9)	0.0213	0.0040	1.06 × 10^−7^	15.3 (14.9–15.6)	0.0492	0.0072	6.94 × 10^−12^ **
Blood pressure and heart rate								
Mean heart rate ^†^ (/min)	35.0 (32.5–38.0)	0.1158	0.0721	0.1079	35.5 (32.5–38.5)	−0.2848	0.1210	0.0186
Systolic BP ^†^ (mmHg)	109.0 (100.5–120.0)	−0.2944	0.1177	0.0124	118.0 (108.7–130.0)	−0.5344	0.2080	0.0102
Diastolic BP ^†^ (mmHg)	67.5 (61.5–74.0)	−0.2066	0.0758	0.0064	73.0 (67.0–80.0)	−0.4107	0.1328	0.0020
Mean BP ^†^ (mmHg)	81.3 (75.0–89.0)	−0.2359	0.0840	0.0050	88.3 (81.3–96.2)	−0.4519	0.1475	0.0022
Lipid profiles								
Total cholesterol ^§^ (mmol/L)	193.0 (171.0–217.0)	0.0007	0.0006	0.2729	199.0 (176.0–222.0)	−0.0001	0.0010	0.9301
HDL cholesterol ^§^ (mmol/L)	59.0 (51.0–68.0)	0.0052	0.0008	3.07 × 10^−12^	51.0 (45.0–59.0)	0.0070	0.0012	4.94 × 10^−9^
LDL cholesterol ^§^ (mmol/L)	115.0 (96.0–136.0)	−0.0008	0.0009	0.3903	126.0 (106.0–146.0)	−0.0004	0.0015	0.7926
Triglyceride ^§^ (mmol/L)	74.0 (55.0–105.0)	−0.0110	0.0016	1.87 × 10^−11^	107.0 (77.0–150.0)	−0.0165	0.0028	6.58 × 10^−9^
Glucose metabolism								
Fasting plasma glucose ^‡^ (mmol/L)	89.0 (85.0–94.0)	−0.1831	0.0952	0.0543	93.0 (88.0–99.0)	−0.5934	0.2276	0.0091
HbA1c ^‡^ (%)	5.5 (5.3–5.8)	−0.0100	0.0038	0.0090	5.7 (5.5–6.0)	−0.0204	0.0090	0.0235
Insulin resistance surrogate markers								
TyG index ^§,‡^	6650.0 (4823.0–9546.0)	−256.27	55.32	4.00 × 10^−6^	9879.0 (6942.0–14,234.5)	−519.90	131.25	7.50 × 10^−5^
TyG-BMI ^§,‡^ (× 10^3^)	143.6 (101.2–211.2)	−5670	1272	8.00 × 10^−6^	274.2 (190.3–404.5)	−14211	3747	1.50 × 10^−4^
TyG-WC ^§,‡^ (× 10^3^)	506.5 (357.2–746.2)	−19,217	4678	4.00 × 10^−5^	887.4 (612.8–1302.6)	−40,422	12118	8.53 × 10^−4^
Atherosclerotic risk factors								
Diabetes mellitus (%)	4.78%	−0.1062	0.0396	0.0074	14.97%	−0.0874	0.0367	0.0173
Hypertension (%)	12.29%	−0.0640	0.0265	0.0159	28.29%	−0.0302	0.0298	0.3112
Metabolic syndrome (%)	9.96%	−0.1254	0.0296	2.23 × 10^−5^	42.30%	−0.1442	0.0269	8.02 × 10^−8^

Abbreviations, adjusted conditions, and subjects recruited for analysis as in Figure 1 and Table 2. *p **—adjusted for age, BMI, and current smoking; Age—adjusted for BMI and current smoking; and BMI—adjusted for age and smoking. Significance was defined as a *p* value of <0.05/(131 + 27) = 3.16 × 10^−4^. *t*-test: ** *p* < 0.001, * *p* < 0.01. ^†^ hypertension, ^‡^ diabetes mellitus, and ^§^ hyperlipidemia.

**Table 4 ijms-23-04165-t004:** Association of the cg08097417 methylation site and age with body shape indices and metabolic traits.

Clinical and Laboratory Parameters	Age (*n* = 1636)					*KLF14* cg08097417 (*n* = 1636)			
	Median (IQR)	Beta	SE	*p* Value *	*p* Value #	Beta	SE	*p* Value *	*p* Value ##
Age (years)	49.0 (40.0–58.0)	-	-	-	-	0.0153	0.0004	7.71 × 10^−243^	
Body shape indices									
Body height (cm)	163.0 (157.5–170.5)	−0.5529	0.0440	1.23 × 10^−34^	2.67 × 10^−17^	−0.0088	0.0010	9.13 × 10^−19^	0.6149
Body weight (kg)	64.6 (55.7–73.5)	−0.6675	0.0527	4.16 × 10^−35^	4.42 × 10^−18^	−0.0104	0.0012	1.88 × 10^−18^	0.7655
Hip circumference (cm)	96.1 (92.0–101.0)	−0.5762	0.0709	8.73 × 10^−16^	4.00 × 10^−6^	−0.0106	0.0016	1.28 × 10^−11^	0.1022
Waist circumference (cm)	83.5 (76.5–90.0)	0.4137	0.0516	2.05 × 10^−15^	1.98 × 10^−7^	0.0068	0.0011	2.30 × 10^−9^	0.5210
Waist–hip ratio	0.86 (0.82–0.91)	66.4649	4.7723	9.62 × 10^−42^	5.31 × 10^−18^	1.1323	0.1062	1.08 × 10^−25^	0.1063
Body fat percentage (%)	26.9 (22.6–32.3)	0.1568	0.0996	0.1155	0.3052	0.0026	0.0022	0.2334	0.8981
Body mass index (kg/m^2^)	23.9 (21.8–26.4)	−0.0126	0.0780	0.8720	0.7680	0.0001	0.0017	0.9349	0.7908
ABSI	7.8 (7.5–8.1)	6.5158	0.5522	6.83 × 10^−31^	3.23 × 10^−15^	0.1040	0.0123	4.64 × 10^−17^	0.5717
WHI	3.9 (3.7–4.1)	14.3854	1.0618	1.05 × 10^−39^	2.62 × 10^−17^	0.2438	0.0236	2.96 × 10^−24^	0.1340
HI	15.1 (14.6–15.6)	−1.0974	0.5654	0.0524	0.8798	−0.0324	0.0123	0.0083	0.0764
Blood pressure and heart rate									
Mean heart rate ^†^ (/min)	34.5 (32.0–37.5)	−0.0671	0.0318	0.0352	0.2659	−0.0013	0.0007	0.0560	0.5506
Systolic BP ^†^ (mmHg)	113.0 (104.0–124.0)	0.2357	0.0184	1.45 × 10^−35^	1.98 × 10^−13^	0.0043	0.0004	7.02 × 10^−25^	0.0126
Diastolic BP ^†^ (mmHg)	71.0 (64.0–79.0)	0.1596	0.0291	5.02 × 10^−8^	0.0087	0.0033	0.0006	2.93 × 10^−7^	0.0610
Mean BP ^†^ (mmHg)	85.0 (78.2–93.3)	0.2432	0.0264	9.40 × 10^−20^	6.21 × 10^−7^	0.0046	0.0006	2.34 × 10^−15^	0.0212
Lipid profiles									
Total cholesterol ^§^ (mmol/L)	193.0 (171.0–217.0)	24.5694	3.4825	2.61 × 10^−12^	3.00 × 10^−5^	0.4394	0.0759	8.53 × 10^−9^	0.2152
HDL cholesterol ^§^ (mmol/L)	53.0 (45.0–63.0)	5.4550	3.0549	0.0744	0.2430	0.0933	0.0663	0.1595	0.8485
LDL cholesterol ^§^ (mmol/L)	120.0 (100.0–141.0)	9.7590	2.3010	2.40 × 10^−5^	0.0211	0.1838	0.0500	0.0002	0.3064
Triglyceride ^§^ (mmol/L)	89.0 (63.0–131.3)	7.2982	1.2616	8.80 × 10^−9^	0.0001	0.1187	0.0275	1.70 × 10^−7^	0.6777
Glucose metabolism									
Fasting plasma glucose ^‡^ (mmol/L)	92.0 (87.0–97.0)	0.0851	0.0173	1.00 × 10^−6^	0.0003	0.0012	0.0004	0.0015	0.7686
HbA1c ^‡^ (%)	5.6 (5.4–5.8)	3.3823	0.4459	5.62 × 10^−14^	1.55 × 10^−9^	0.0449	0.0098	5.00 × 10^−6^	0.3136
Insulin resistance surrogate markers									
TyG index ^§,‡^	8239.0 (5670.0–12,240.0)	8.20 × 10^−5^	2.70 × 10^−5^	0.0026	0.1158	2.00 × 10^−6^	5.98 × 10^−7^	0.0070	0.4128
TyG-BMI ^§,‡^ (×10^3^)	199.3 (127.5–314.5)	2.00 × 10^−6^	1.00 × 10^−6^	0.0187	0.2553	4.89 × 10^−8^	2.22 × 10^−8^	0.0280	0.4433
TyG-WC ^§,‡^ (×10^3^)	693.3 (440.5–1069.5)	8.56 × 10^−7^	2.98 × 10^−7^	0.0042	0.1554	1.72 × 10^−8^	6.52 × 10^−9^	0.0083	0.3771
Atherosclerotic risk factors									
Diabetes mellitus (%)	4.83%	7.7733	1.2499	6.33 × 10^−10^	5.00 × 10^−6^	2.2274	0.5205	1.90 × 10^−5^	0.9187
Hypertension (%)	18.95%	7.8538	0.6828	1.7 × 10^−29^	2.58 × 10^−10^	3.0710	0.3183	4.99 × 10^−22^	0.0041
Metabolic syndrome (%)	18.89%	4.2314	0.7435	1.49 × 10^−28^	3.10 × 10^−5^	1.3859	0.3097	8.00 × 10^−6^	0.9530

Abbreviations, adjusted condition, and subjects recruited for analysis as in Figure 1 and Table 2. ** p*—adjusted for age, sex, BMI, and current smoking. # *p*—adjusted for age, sex, BMI, current smoking, and cg08097417; ## *p*—adjusted for age, sex, BMI, current smoking, and age. Significance between age and studied phenotypes was defined as a *p* value of <0.05/(24 + 26) = 0.001. Significance between cg08097417 and studied phenotypes was defined as a *p* value of <0.05/(24 + 27) = 9.80 × 10^−4.^. ^†^ hypertension, ^‡^ diabetes mellitus, and ^§^ hyperlipidemia.

## Data Availability

The data presented in this study are available on request from the corresponding author.

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
