# Peer review of "Differential Genetic and Epigenetic Effects of the KLF14 Gene on Body Shape Indices and Metabolic Traits"

_ijms, 2022, doi:10.3390/ijms23084165_

Round 1

Reviewer 1 Report

The manuscript presented by Semon Wu et al. entitled “Differential genetic and epigenetic effects of the KLF14 gene on body shape indices and metabolic traits” is of interest and finally proves the epigenetic involvement in body shape, which include cg08097417 methylation on the KFL14 gene.

I suggest the authors include in the discussion more details concerning also microRNAs influence in the body shape please consider the following reference Microrna. 2019;8(2):116-126. doi: 10.2174/2211536608666181126093903 and take off dot at the end of the  conclusion section

Author Response

Ans: Thank you for your suggestion. We have cited as recommendation and included “Interestingly, Cannataro et al.,[59] found that microRNAs may modify promoter by methylation on targeted genes and regulate its expression, which affect body shape under ketogenic diet. Therefore, a study to investigate the role of microRNA candidates in regulating KLF14 promoter methylation may help to elucidate more of the mechanism of KLF14 associated body shape changes.” in the Discussion section.

Reviewer 2 Report

I read with interest the article by  Wu et al. The work is elegant well-designed and well written with a large number of samples.

My recommendation is to accept.

Author Response

Ans: Thank you for your comment.

Reviewer 3 Report

In this manuscript, Wu et al reported the association of the DNA methylation of KLF14 genes with various phenotypes in 1636 participants.

Major:

  1. The authors need to provide negative controls. For example, showing other genes don’t associate with these phenotypes. Without proper negative controls, it is hard to determine the DNA methylation levels of this KFL1 gene are truly biologically associated with the traits as the author claimed in the tested ~1k populations. It is could be just aging (common phenomenon for a lot of genes).
  2. Regarding the design of the assay. The authors filtered a lot of participants’ data and only the very healthy participants are selected for methylation studies. So, if all the participants are healthy, all the phenotypes are within the normal range, then how does this gene relate to health? What is the point of this study?

Minor:

  1. Bisulfite treatment converts unmethylated C to T, for the C/T, T/C mutations listed in Table 1, how are these mutations processed? Along the same line, how many CpG sites around and in KLF14? Can Infinium chip detect them?
  2. Table 2, the phenotypes need to be grouped by gender (age, body height etc).
  3. Figure 2, panel a. In the legend, the authors stated “(A) Association between KLF14 promoter methylation cg08097417 level and chronologic age”. How is the promoter defined for this gene? Negative control needed. Is it possible that the methylation levels of other genes are correlated with age too, maybe it is just a common phenomenon of aging?
  4. Table 3 and 4, the phenotypes need to be provided as the same way in Table 2.
  5. Figure 3 and 4, need to provide DNA methylation levels.
  6. Figure 5, how is “standardized effect sizes” defined, need to provide more description in the method section.

Author Response

Major:

Question 1: The authors need to provide negative controls. For example, showing other genes don’t associate with these phenotypes. Without proper negative controls, it is hard to determine the DNA methylation levels of this KFL14 gene are truly biologically associated with the traits as the author claimed in the tested ~1k populations. It could be just aging (common phenomenon for a lot of genes).

Ans: Thank you for your comment. We believed that other genes may be associated with these phenotypes. In other words, it is likely that many genes may be associated with these phenotypes but not with age, suggesting different mechanisms underlying the associated pathways. From our statistic results, we demonstrated KLF14 are associated with metabolically healthy status, gender, and age at genetic and epigenetic level. Hopefully, our study in the future may identify more underlying mechanism that associated with same metabolic phenotype but age-independent.

Question 2: Regarding the design of the assay. The authors filtered a lot of participants’ data and only the very healthy participants are selected for methylation studies. So, if all the participants are healthy, all the phenotypes are within the normal range, then how does this gene relate to health? What is the point of this study?

Ans: Thank you for your comment. We agree that our study population belongs to a relatively healthy population, which excluded cancer history and other acute symptoms. However, the studied participants are not illness free and many of them show chronic diseases instead, including hypertension and diabetes mellitus. Therefore, it reflects to general health situation in Taiwan or in other Asian populations. In addition, we believe that the genetic and epigenetic patterns on KLF14, such as age-, sex- and obesity-dependent modifications may also be different from other ethnic populations.

Minor:

Question 1: Bisulfite treatment converts unmethylated C to T, for the C/T, T/C mutations listed in Table 1, how are these mutations processed? Along the same line, how many CpG sites around and in KLF14? Can Infinium chip detect them?

Ans: Thank you for your comment. The C/T, T/C mutations were performed by genotyping with Axiom Genome-Wide CHB 1 or 2 Array. The DNA samples do not treat with sodium-bisulfite, so it cannot be detected with Infinium MethylationEPIC BeadChipEPIC array. There are 32 CpG sites on KLF14 that were listed in Supplementary Table 4. These DNA samples need to be treated with sodium-bisulfite and detected byInfinium MethylationEPIC BeadChipEPIC array.

Question 2: Table 2, the phenotypes need to be grouped by gender (age, body height etc).

Ans: Thank you for your comment. We added the median (IQR) or percentage of each phenotype in Table 2 by gender.

Question 3: Figure 2, panel a. In the legend, the authors stated “(A) Association between KLF14 promoter methylation cg08097417 level and chronologic age”. How is the promoter defined for this gene? Negative control needed. Is it possible that the methylation levels of other genes are correlated with age too, maybe it is just a common phenomenon of aging?

Ans: Thank you for your comment. We defined the promoter region from the PUBMED DNA database and the related region of KLF14 also listed in Supplementary Table 4. As has been previously reported, many genes have been closely related to age with aging-related alteration of hypermethylation in whole blood samples, thus providing an accurate estimate of chronological age and serving as age-related epigenetic biomarker. It is possible that the methylation levels of other genes are correlated with age too, maybe it is just a common phenomenon of aging. We hoped that further study may help to elucidate the problem.

Question 4: Table 3 and 4, the phenotypes need to be provided as the same way in Table 2.

Ans: Thank you for your comment. We add the median (IQR) or percentage of each phenotype in Table 3 and 4.

Question 5: Figure 3 and 4, need to provide DNA methylation levels.

Ans: Thank you for your comment. We provide the DNA methylation levels in Supplementary Table 4 instead of in Figure 3 and 4.

Question 6: Figure 5, how is “standardized effect sizes” defined, need to provide more description in the method section.

Ans: Thank you for your comment. To be more accurate, we have corrected the “standardized effect sizes” to the” standardized coefficients (beta)” in figure legend of Figure 5. We also revised and added “The Radar plots were presented with the standardized coefficients (beta) in the linear regression.” in the method section.

Round 2

Reviewer 3 Report

The authors have addressed my concerns.